

# A new clustering method based on multipartite networks

Rodica-Ioana Lung

Center for the Study of Complexity, Babes-Bolyai University of Cluj-Napoca, Cluj Napoca, Cluj, Romania

## ABSTRACT

The clustering problem is one of the most studied and challenging in machine learning, as it attempts to identify similarities within data without any prior knowledge. Among modern clustering algorithms, the network-based ones are some of the most popular. Most of them convert the data into a graph in which instances of the data represent the nodes and a similarity measure is used to add edges. This article proposes a novel approach that uses a multipartite network in which layers correspond to attributes of the data and nodes represent intervals for the data. Clusters are intuitively constructed based on the information provided by the paths in the network. Numerical experiments performed on synthetic and real-world benchmarks are used to illustrate the performance of the approach. As a real application, the method is used to group countries based on health, nutrition, and population information from the World Bank database. The results indicate that the proposed method is comparable in performance with some of the state-of-the-art clustering methods, outperforming them for some data sets.

## INTRODUCTION

Clustering methods aim to identify groups of similar instances of data to reveal common characteristics that are not visible by other means. They represent one of the major classes of machine learning methods with multiple practical applications (*Ezugwu et al., 2022*) as they reveal connections within the data without using supplementary information. There are many criteria for their classification, but recent trends group them into two major categories: traditional and modern (*Xu & Tian, 2015*; *Anand & Kumar, 2022*). Traditional methods are textbook approaches that tend to be used more by practitioners in other research fields for practical applications. While they may be outperformed by some newer approaches for some data sets, they have also passed the test of time, as they offer consistent and competitive results that are interpretable and reliable. They are also relatively easy to use and available in various implementations. Modern approaches tend to use newly developed concepts such as deep neural networks or graph theory. A separate trend is to propose general improvement methods, *e.g.*, methods that can enhance the results of the clustering by ameliorating the data set (*Li et al., 2023*; *Wang et al., 2018*).

The fact that the market for new clustering methods is saturated with variations of traditional and modern methods proposed either as stand-alone clustering algorithms or

Corresponding author
Rodica-Ioana Lung,
rodica.lung@econ.ubbcluj.ro

as solutions for particular practical applications using specific information related to the data should not be seen as dismissing the exploration of new approaches. An attempt at a new approach is proposed in this article: the Multipartite Network Clustering (MN-C) algorithm. MN-C uses a multipartite network to identify clusters in the data. The layers of the multipartite network correspond to attributes of the data. The nodes of each layer represent intervals for the corresponding attribute and are populated with instances having the value of this attribute within that interval. Edges of the network connect the nodes that contain the attributes of the same data instance. The clusters approximately correspond to paths in the network. Numerical experiments illustrate the behaviour of the approach on a set of synthetic and real-world benchmarks.

## METHODS

Most clustering methods approach the problem in one of the following manners: using a representation for the clusters in the form of central points or distribution in representative or partition-based clustering; constructing clusters by successively aggregating/dividing data in hierarchical clustering; using data density to define clusters (*Bhattacharjee & Mitra, 2020*). Graph-based models, in which the data is converted into a graph, and spectral models, closely connected to the latter, in which matrix representations of graphs are further analysed, are also popular (*Hloch, Kubek & Unger, 2022*). There are many variants of these main approaches, using concepts from related fields, such as fuzzy computing or neural networks (*Ayyub et al., 2022*).

The standard versions of the approaches mentioned above are considered traditional methods (*Xu & Tian, 2015*). They include variants of partition-based methods, such as k-means, partitioning around medoids (PAM), or Clustering Large Applications (CLARA). DBSCAN and its variants are density-based methods in this group. Other traditional methods are hierarchical trees, expectation–maximization clustering, and fuzzy analysis clustering. Most of them can be found in textbooks (*Zaki & Meira Jr., 2014*) and have implementations available for free or within commercial data analysis software packages.

Modern methods extend the traditional ones by including techniques from related fields, such as network analysis or deep learning (*Wierzchoń & Kłopotek, 2017*). They include the class of spectral clustering methods (*Nascimento & de Carvalho, 2011*), affinity propagation clustering (*Frey & Dueck, 2007*), density peaks clustering methods (*Hou, Zhang & Qi, 2020*), and various deep clustering methods (*Anand & Kumar, 2022*; *Zhou et al., 2022*). Advanced methods using graphs aim to perform the clustering on graphs, such as graph neural networks (*Tsitsulin et al., 2020*), a marginalized graph autoencoder in *Wang et al. (2017)* and a local high-order graph clustering in *Yin et al. (2017)*.

The present article presents a network-based clustering method that identifies clusters from a multipartite network constructed from attributes of the data. In what follows, other clustering methods that use network techniques are succinctly reviewed, and the newly proposed method, Multipartite Network Clustering (MN-C), is introduced.

## Related work

Let $\mathbf{X} \in \mathbb{R}^{n \times d}$ be a data set containing $n$ instances from $\mathbb{R}^d$ and with $d$ attributes/features. We denote by $\mathbf{X}_j$ the $n$-dimensional vector corresponding to attribute $j$, $j = 1, \ldots, d$. The clustering problem consists of grouping instances based on some similarity indicator. While it is ideal for a method to determine the number of clusters in the data, sometimes, for specific applications, the desired number of clusters $k$ is indicated or required.

Most graph-based clustering methods construct a graph by treating the instances of the data as the nodes in the graph. A similarity measure is used to compute the weight of the edge connecting two nodes. Similar nodes will form cliques or communities that can be detected using various tools, thus revealing the clusters in the data. Spectral methods analyse matrix representations of the graph to extract information about communities (*Washio & Motoda, 2003*; *Foggia et al., 2009*). A discussion of the importance of constructing the graph and suggested solutions can be found in *Nie et al. (2016)*; *Maier, Luxburg & Hein (2008)*. In *Huang et al. (2020)*, an ultra-scalable spectral clustering algorithm, called UNSPEC, designed for extremely large-scale data sets with limited resources is presented.

There are many applications of graph-based methods: in text clustering (*Hloch, Kubek & Unger, 2022*) and text representations (*Rao & Chakraborty, 2021*), in anomaly detection (*Akoglu, Tong & Koutra, 2015*), labeling crime data (*Das & Das, 2019*), image search (*Yan et al., 2017*), clustering protein sequences (*Kawaji, Takenaka & Matsuda, 2004*), *etc.*

## Multipartite network clustering (MN-C)

Multipartite-Network Clustering clusters the data by using a multipartite graph constructed from the data set. The network layers correspond to attributes of the data, and we assume that we are searching for $k$ classes. The main components of the MN-C algorithm are described in detail in what follows.

### *The multipartite graph*

A multipartite graph (*Van Dam, Koolen & Tanaka, 2016*) is a graph whose nodes can be divided into disjoint sets, which are also called layers, within which no two nodes are adjacent. A $d$-partite graph contains $d$ layers.

MN-C uses a multigraph denoted by $G(V, E | \mathbf{X})$, where $V$ is the set of nodes and $E$ is the set of edges constructed from the data set $X$. The number of layers of $G$ is equal to the number of attributes of $\mathbf{X}$, denoted by $d$ in this article. Each layer has a maximum of $k$ nodes, where $k$ is the number of classes that we are searching for.

### *Layers and nodes*

Each layer of the graph corresponds to an attribute in the data set. Each attribute $\mathbf{X}_j$, $j = 1, \ldots, d$, is divided into $k$ intervals denoted by $I_{jl}$, $l = 1, \ldots, k$ and a network node $V_{j,l}$ is created for each interval. The way intervals are computed defines the structure of the network. They can be designed in various manners, *e.g.*, of equal length, to contain an equal number of instances, or by including some problem-dependent information. In this approach, MN-C constructs intervals of equal lengths between $\min(\mathbf{X}_j)$ and $\max(\mathbf{X}_j)$.

The network has $d$ layers and a maximum of $d \times k$ nodes. Each node 'contains' a number of instances. An instance $x_i \in \mathbf{X}$ will be placed in a node in each layer of the network, corresponding to the interval within each of its components belongs:

$$x_i \in V_{j,l} \quad \text{if } x_{ij} \in I_{jl}.$$

---

**Algorithm 1** Construction of Multipartite Network

---

1: **input:** data set $\mathbf{X}$, number of classes $k$;
2: **output:** network $G = (V, E | \mathbf{X})$;
3: $V = \emptyset$;
4: **for** each attribute $j$ **do**
5:     Compute $k$ intervals $I_{jl}, l = 1, \ldots, k$ {of equal size} between $\min(\mathbf{X}_j)$ and $\max(\mathbf{X}_j)$;
6:     Add node $v_{jl}$ to $V$, where $j$ is the layer of the node
7: **end for**
8: **for** each data instance $x_i \in \mathbf{X}$ **do**
9:     Create path $v_{1l_1}, v_{2l_2}, \ldots, v_{dl_d}$ with $x_{ij} \in I_{jl_j}$ $\forall$j=1,\ldots,$d-1$, by adding edges $(v_{jl_j}, v_{j+1,l_{j+1}})$ to $E$; if an edge already exists, increase its weight by 1;
10: **end for**
11: Remove empty nodes from $V$;
12: **return** $d$-partite graph $G = (V, E | \mathbf{X})$, and $I = \{I_{jl}\}_{j=\overline{1,d}, l \in \{1,\ldots,k\}}$ intervals corresponding to nodes;

---

### Edges

Each instance $x_i$ connects the nodes to which its attributes belong. Thus, if instance $x_i$ is placed in nodes $V_{j,l'}$ and $V_{j+1,l''}$, for $j = 1, \ldots, d-1$, an edge is added between the two nodes if it does not yet exists. If an edge already exists between the two nodes, its weight is increased by one. Thus, each instance in the data creates a path between the first layer of the network and the last one.

The weight of an edge represents the number of instances that are placed in the connected nodes, *i.e.,* belonging to the same corresponding intervals. The number of instances placed in each interval will vary depending on their distribution. Intervals with no instances will be ignored, *i.e.,* empty nodes will be removed from the network.

**Example 1** Consider a data set with 100 instances, six attributes, four clusters that are well separated (generated by using the `make_classification` function from the `scikit-learn` package in `Python` (*Pedregosa et al., 2011*), using for `class_sep=10`, resulting in four well-separated clusters). Figure 1 presents the corresponding multipartite network with six layers, each having two or three nodes.

### Initial clusters

Each instance of the data set represents a network path from the first layer to the last one. It is reasonable to assume that instances representing the same path may belong to the same

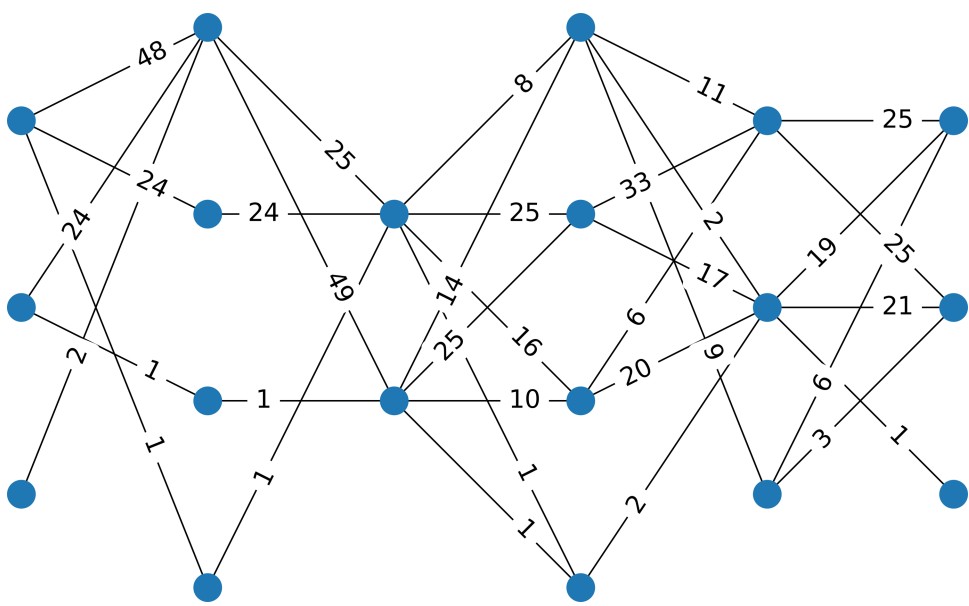

**Figure 1  Example 1** The network corresponds to a data set having 100 instances, six attributes, and four classes. The data are well separated. The weight of an edge indicates the number of instances that have components in the two intervals represented by the nodes.

cluster. In the first step of MN-C, instances that share a common path throughout all the layers of the network are placed in the same cluster.

Depending on the structure of the data, this procedure may result in any number of clusters, which may be greater than $k$, due to the number of possible paths in the network. Not all paths in the network represent instances. We denote by $k_0$ the number of clusters resulting from the this initialization of the clustering. In what follows, a procedure to merge clusters in order to reduce their number is performed, if necessary, *i.e.*, if $k_0 > k$.

### Merging clusters

A cluster $C$ represents a path starting from the first layer to the last layer of the network $G$. In the initial stage, each cluster contains one path. Clusters are merged by finding paths that have the most common elements. Merging is performed in the order of the number of instances corresponding to each cluster, starting with the smallest ones. The size of the cluster is denoted by $\mu(C)$ and is the number of instances assigned to it. In the initial stage, it represents the number of instances that belong to all node intervals from the cluster.

It is assumed that larger clusters are more stable, while those containing fewer instances should be merged. In order to find, for each cluster $C$, the cluster for it to be merged with, we compute the following strength indicator for two clusters $C$ and $C'$:

$$S(C,C') = \sum_{e_{i,i+1} \in C,C'} w_{ij}. \tag{1}$$

Thus, $S$ adds the weights of common edges between two clusters. Each cluster is merged with the one for which the strength indicator is the highest, *i.e.*, they have more edges/instances

in common. Thus

$$C^* = \underset{C'}{\operatorname{argmax}} S(C, C'). \tag{2}$$

Clusters $C$ and $C^*$ are merged by placing all instances from $C$ in $C^*$. All clusters are merged except the $k$ largest ones. This process reduces the number of clusters from $k_0$ to $k_1$. To reach the desired number of clusters $k$, the process is repeated for several iterations while the number of clusters is greater than $k$, and stops when it reaches it, or gets smaller than $k$.

**Remark 1** If $S(C, C') = 0$ for all $C' \neq C$, then the merging cluster is found by looking for the cluster having the most nodes in common with $C$. We write $v(C, C') = |v \in V | v \in C \wedge v \in C'|$. If there is no such cluster, then $C$ is not merged with any other cluster.

### *Outcome*

The outcome of MN-C is a set of clusters, corresponding to network paths. Each instance in the data is assigned to a cluster.

---

**Algorithm 2** MN-C algorithm

---

1:  **Input:** Data set $\mathbf{X} \in \mathbb{R}^{n \times d}$, number of clusters $k$
2:  **Output:** Set of clusters $C_1, C_2, \ldots, C_k$;
3:  Construct weighted multipartite network $G = (V, E)$ (Algorithm 1)
4:  Assign all data instances belonging to the same network path from the first to last layer to the same cluster forming $k_0$ clusters;
5:  $it = 0$;
6:  $Change = True$;
7:  **while** $k_{it} > k$ **and** $Change$ **do**
8:      Order clusters in ascending order of size $\mu(C)$;
9:      **for** each cluster $C_l, l=1, \ldots, k_{it} - k$ **do**
10:         Set $C_l^* = \operatorname{argmax}_{z=l+1, k_{it}} S(C_l, C_z)$;
11:         **if** $S(C_l, C_l^*) \neq 0$ **then**
12:             Set $C_l = C_l^*$;
13:         **else**
14:             Set $C_l^* = \operatorname{argmax}_{z=l+1, k_{it}} |v(C_l, C_z)|$;
15:             **if** $|C_l^*| \neq 0$ **then**
16:                 Set $C_l = C_l^*$;
17:             **end if**
18:         **end if**
19:     **end for**
20:     If no change has been made to clusters, $Change = False$
21:     $it \leftarrow it + 1$;
22: **end while**
23: **return** Clusters $C_1, C_2, \ldots, C_{k_{it-1}}$

---

# NUMERICAL EXPERIMENTS

The performance of MN-C is evaluated on a set of synthetically generated and real-world data sets for clustering and classification and compared with that of other standard state-of-the-art clustering models.

## Experimental set-up

In this subsection, the data used for the experiments is described, as well as the methods used for the comparisons and the performance metric used for evaluating the results.

### Synthetic data sets

In order to illustrate the behaviour of MN-C on different types of data, a set of synthetic data sets with various characteristics was generated by using the `make_classification` (https://scikit-learn.org/stable/modules/generated/sklearn.datasets.make_classification.html, accessed Jan 2023) function in `Python`, by combining the following parameters:

- number of instances: 100, 200, 500, 1000, 2000;
- number of attributes: 30, 50, 100, 150, 200, 250, 1000;
- class separator: 0.1, 0.5, 1, controlling the overlap of different clusters;
- number of classes: 30, 50, 100, 150, 200, 250, 1000.

All reasonable combinations of parameter were considered, *i.e.,* excluding the settings with more classes than the number of instances.

### Real-world data sets

A selection of data sets used for clustering and classification from the UCI Machine learning repository (*Dua & Graff, 2017*) is presented. The names and characteristics of the data are listed in Table 1.

### Comparisons with other methods

MN-C is compared with 4 clustering methods that aim to find a given number of clusters: Kmeans, Gaussian Mixture (GM), Affinity Propagation (AP), Birch, and *Zaki & Meira Jr. (2014)*; *Frey & Dueck (2007)*; *Zhang, Ramakrishnan & Livny (1996)*. Their corresponding implementation in the `sklearn` Python library (*Pedregosa et al., 2011*) is used with its default parameters. In addition, for the real-data sets, the results are also compared to two spectral clustering methods, the standard implementation in `Python`, which we will call SC and the Ultra-Scalable Spectral Clustering (UNSPEC), using the code provided by the authors (*Huang et al., 2020*).

### Performance evaluation

In order to evaluate and compare the results, the normalized mutual information indicator (NMI) is used (*Zaki & Meira Jr., 2014*). NMI takes values between 0 and 1 and can be used to compare results provided by a clustering method with a baseline represented by the known clusters. Values closer to 1 indicate a better match.

**Table 1**  UCI Machine learning repository data sets used to illustrate the behaviour of MN-C.

| Data | Name | Instances | Attributes | Clusters |
|------|------|-----------|------------|----------|
| R1 | Mammographic Mass | 961 | 5 | 2 |
| R2 | Statlog (Vehicle Silhouettes) | 846 | 18 | 4 |
| R3 | Breast Cancer Wisconsin (Prognostic) | 198 | 33 | 23 |
| R4 | Thyroid allbp | 2800 | 26 | 5 |
| R5 | Statlog (Australian Credit Approval) | 690 | 14 | 2 |
| R6 | Acute Inflammations Data Set | 120 | 6 | 2 |
| R7 | Turkiye Student Evaluation Data Set | 5820 | 32 | 13 |
| R8 | Bank Marketing | 4521 | 16 | 2 |
| R9 | BLOGGER | 100 | 5 | 2 |
| R10 | Cardiotocography 10 class | 2126 | 35 | 10 |
| R11 | Dermatology | 366 | 34 | 6 |
| R12 | Flags - color | 194 | 29 | 8 |
| R13 | Flags - religion | 194 | 29 | 8 |
| R14 | Hungarian Heart Disease | 294 | 13 | 5 |
| R15 | Leaf | 340 | 15 | 30 |
| R16 | Lenses | 24 | 4 | 3 |

## Results and discussion

### Synthetic data sets

We generated 588 synthetic data sets. The results are summarised in Fig. 2 and Table 2. Figure 2 presents scatter plots of the values of the NMI obtained by MN-C and each of the other methods. It is used to illustrate the overall performance of MN-C compared to the other method: points located above the first bisector (represented in each image by a grey dashed line) indicate that MN-C performs better. The points located below it indicate that the NMI obtained by the other method are better. For each method, the plot is separated based on the values of the class separator parameter, as this is the one that controls the difficulty of the clustering problem.

Table 2 summarises the results obtained on the synthetic data sets in the following manner: for each characteristic of the data set, the percentage of results for which MN-C obtained higher NMI than the other method is presented. Next to each number, an asterisk (*) indicates if, considering all data sets with the same characteristic, a $t$-test comparing MN-C's values of the NMI finds them to be significantly greater than those obtained by the other method. A minus sign (-) indicates that overall there is no significant difference between the methods. An (x) indicates that the results obtained by the other method are significantly better.

The results indicate that MN-C is a competitive method, providing better results than Kmeans, Gaussian Mixture, and Affinity propagation, but not better than the Birch method, for the synthetic data sets. However, Fig. 2 shows that while the results obtained by MN-C are worse in most cases, they are close to those obtained by Birch as they align to the first bisector. We find in general MN-C better than the other three methods in more instances

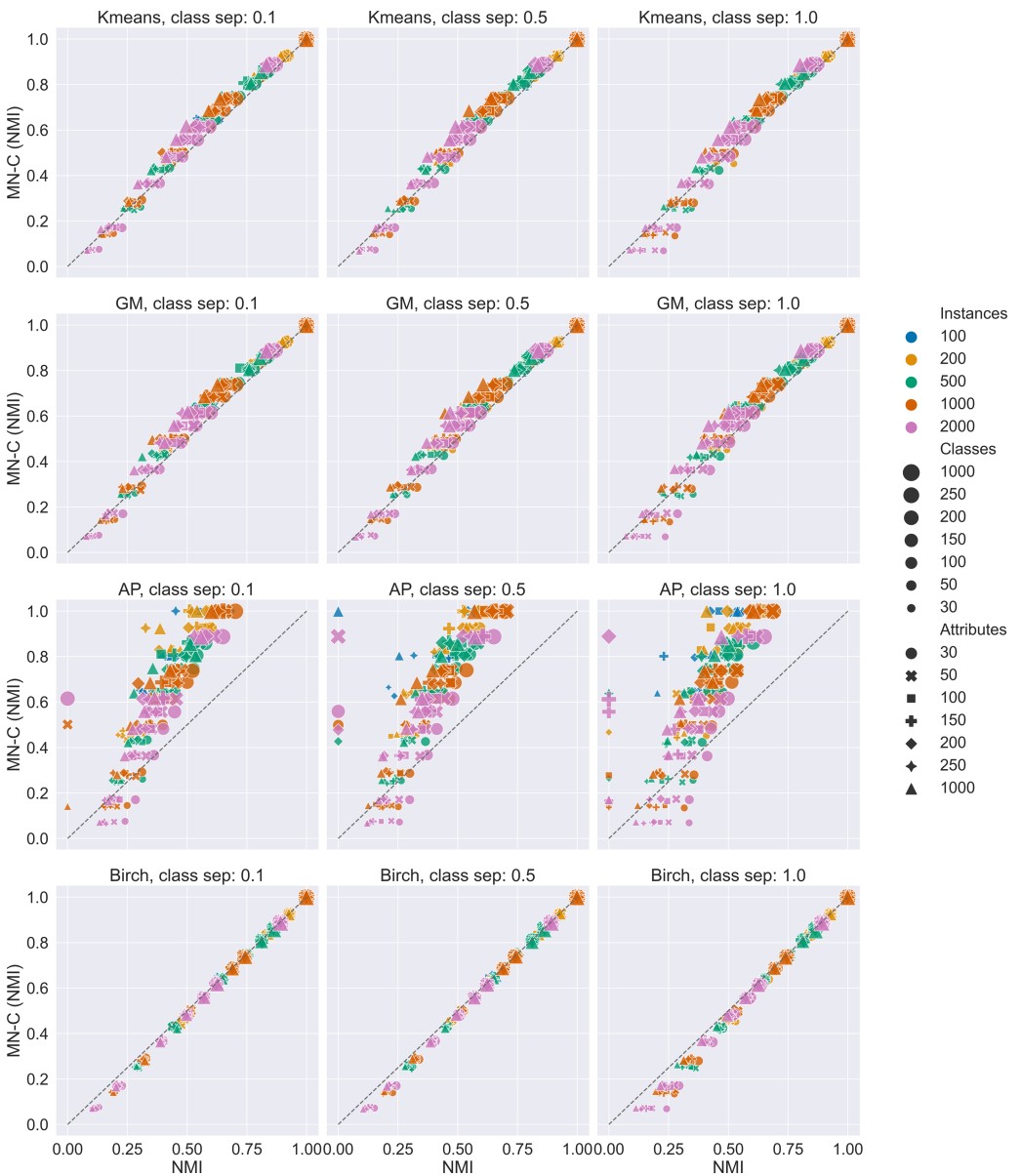

**Figure 2 Overview of results for the synthetic data sets.** Scatter plots of NMI values obtained by each method compared with MN-C. A point above the first bisector represented as a dashed line indicates that the NMI obtained by MN-C for the data set is greater than that obtained by the other method. Compared with Kmeans 84.01% of MN-C results are above the line; with GM, 86.39% of results are above the line; with Birch, 17.68% of results are above the line; and with AP, 88.94% of results are above the line.

for the most difficult settings, for example for class separator values of 0.1, a smaller number of instances, a large number of attributes, and a larger number of classes.

### Varying the number of clusters

MN-C uses the number of clusters as a parameter. For different values of this parameter, it will divide the data into clusters that also contain grouped data, in a manner similar to that

**Table 2** Overview of results obtained on synthetic data sets: percentage of data instances in which MN-C's NMI are greater than that of the other method for different characteristics of the data sets. Detailed values are presented in Table 3.

| Class sep. | Kmeans | GM | Birch | AP |
|---|---|---|---|---|
| 0.1 | 86.73[*] | 88.27[*] | 19.39 × | 88.78[*] |
| 0.5 | 83.67[*] | 87.76[*] | 19.9 × | 88.27[*] |
| 1 | 81.63[*] | 83.16[*] | 13.78 × | 89.8[*] |
| Instances | Kmeans | GM | Birch | AP |
| 100 | 96.83[*] | 100.0[*] | 55.56 × | 100.0[*] |
| 200 | 95.24[*] | 97.14[*] | 35.24 × | 100.0[*] |
| 500 | 88.89[*] | 90.48[*] | 7.14 × | 96.83[*] |
| 1000 | 80.27[*] | 82.99[*] | 15.65 × | 85.71[*] |
| 2000 | 70.07[*] | 72.79[*] | 0.0 × | 72.79[*] |
| Attributes | Kmeans | GM | Birch | AP |
| 30 | 67.86 - | 70.24 - | 16.67 × | 79.76[*] |
| 50 | 76.19[*] | 77.38[*] | 19.05 × | 86.9[*] |
| 100 | 84.52[*] | 88.1[*] | 21.43 × | 89.29[*] |
| 150 | 84.52[*] | 89.29[*] | 17.86 × | 90.48[*] |
| 200 | 89.29[*] | 92.86[*] | 13.1 × | 89.29[*] |
| 250 | 91.67[*] | 90.48[*] | 15.48 × | 92.86[*] |
| 1000 | 94.05[*] | 96.43[*] | 20.24 × | 94.05[*] |
| Classes | Kmeans | GM | Birch | AP |
| 30 | 42.86 - | 52.38 - | 9.52 × | 60.95[*] |
| 50 | 79.05[*] | 79.05[*] | 10.48 × | 79.05[*] |
| 100 | 92.38[*] | 94.29[*] | 22.86 × | 98.1[*] |
| 150 | 96.43[*] | 98.81[*] | 14.29 × | 100.0[*] |
| 200 | 98.81[*] | 98.81[*] | 26.19 × | 100.0[*] |
| 250 | 100.0[*] | 100.0[*] | 6.35 × | 100.0[*] |
| 1000 | 100.0[*] | 100.0[*] | 50.0 × | 100.0[*] |

**Notes.**

[*]An asterisk (*) indicates that overall results for that particular data set characteristic obtained by MN-C are significantly better than the other method.

[-]A minus sign (-) indicates that there is no significant difference, and a (×) indicates that results obtained by the other method are better.

of the other clustering methods. For example, given a synthetic data set with 100 instances, 150 attributes, 30 classes, and a class separation parameter of 0.1, the NMI obtained by all the methods, except AP which determines the number of clusters on its own, have a similar trend, as illustrated in Fig. 3. The increasing values of NMI for all the methods indicate that when more clusters are obtained, instances that belong to the same clusters are grouped together in sub-clusters. Since NMI is an external performance measure it cannot be used to detect the number of clusters, but it can show that the performance of the method may be considered robust with respect to this parameter, as it places instances from the same cluster together. It may also indicate that any method of determining the number of clusters in the data based on some internal quality measure, such as the elbow method (*Yager & Filev, 1994*; *Thorndike, 1953*), or information-based methods (*Sugar & James, 2003*), can be used with MN-C in a manner similar to how they are used with other

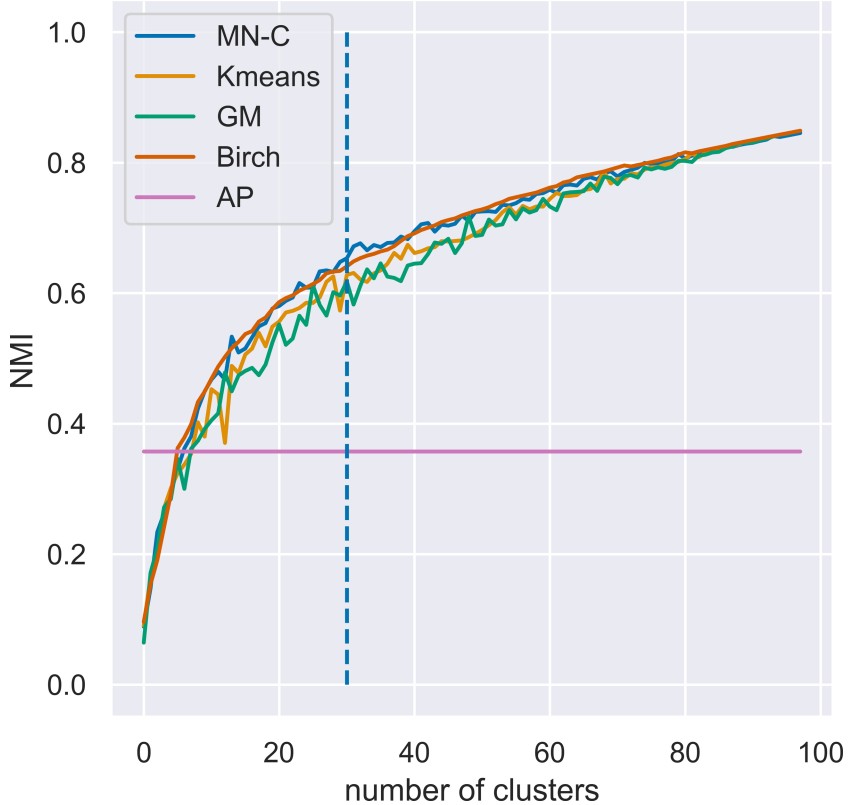

**Figure 3** Evolution of values of NMI for different numbers of clusters *k* for a synthetic data set with 100 instances, 150 attributes, 30 classes, and a class separation parameter of 0.1. The vertical line indicates the number of clusters in the data set.

clustering methods (*Fu & Perry, 2020*). Figure 4 illustrates the values of three internal indicators that are known to be used with an elbow method to evaluate the number of clusters based on the results of a learner for the same data set as in Fig. 3. The silhouette score, distortion, and inertia all have descending values; a linear trend may be observed at their left.

### Real-world benchmarks

Results obtained by the four methods on the real-world benchmarks are presented in Table 4. The value of the NMI and number of clusters determined by each method are indicated. The data sets vary in the number of clusters, instances, and attributes. We show a variety of situations in which the results obtained by MN-C are better than those obtained by the other methods (including Birch, unlike the case of the synthetic data sets). The results also illustrate the reality of the variability in clustering performance with large differences between the methods for some data sets, *e.g.*, R6, for which MN-C obtains a value of 0.662 for the NMI, while the other methods obtain 0.041, 0.026, 0.453, and 0.031, respectively. AP obtains 0.453 but with 11 clusters instead of two. This situation arises also

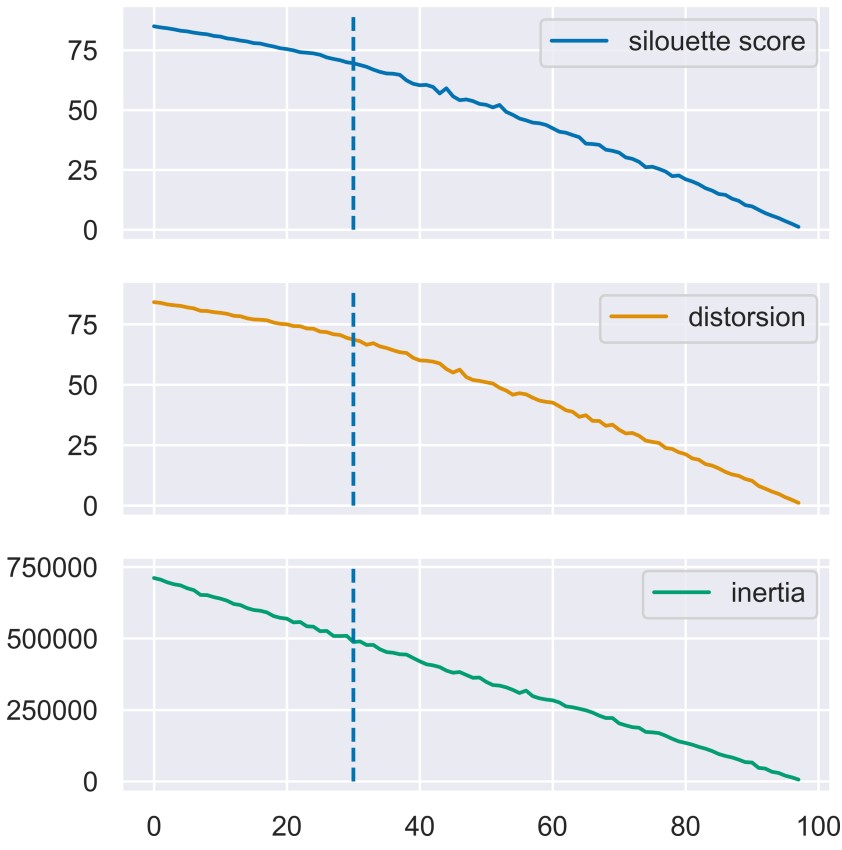

**Figure 4 Three internal performance indicators for different numbers of clusters, the same data set as in Fig. 3.** The vertical lines indicate the number of clusters in the data set. An almost linear trend can be noticed at the left of this line.

for other data sets. While these results cannot be generalized, they do indicate the potential of MN-C to undercover the clustering structure of different real-world data sets.

## A real world application

A country's health and nutrition situation is often assumed to be directly linked with economic indicators. Each year, the World Bank categorizes countries into four income groups: Low income, Lower middle income, Upper middle income, and High income based, on GNI per capita.

GNI per capita stands for Gross National Income per capita. It is a measure used to assess the economic well-being of a country and its residents. GNI represents the total income a country's residents earn, including domestically and income generated abroad. 'Per capita' means that the GNI is divided by the country's total population, giving an average income figure for each individual. GNI per capita is often used as an indicator to compare the average income levels between different countries or to track a single country's economic growth and development over time. It provides a useful metric for understanding the average income and standard of living of the population in a particular nation. Figure 5 presents a map of the distribution of countries into the four groups in 2022.

**Table 3** Average and standard deviation for NMI obtained by all the methods on the synthetic data sets, grouped by one characteristic of the data: the class separator, number of instances, number of attributes, and number of classes, from which the data obtained in Table 2 are computed. Table 2 also indicates significant results comparing values.

| | MN-C | | Kmeans | | GM | | AP | | Birch | |
|---|---|---|---|---|---|---|---|---|---|---|
| | mean | std | mean | std | mean | std | mean | std | mean | std |
| Class sep. | | | | | | | | | | |
| 0.1 | 0.6120 | 0.2644 | 0.5801 | 0.2588 | 0.5728 | 0.2602 | 0.3845 | 0.1421 | 0.6254 | 0.2505 |
| 0.5 | 0.6120 | 0.2641 | 0.5808 | 0.2559 | 0.5736 | 0.2587 | 0.3755 | 0.1511 | 0.6273 | 0.2484 |
| 1.0 | 0.6118 | 0.2644 | 0.5901 | 0.2486 | 0.5834 | 0.2522 | 0.3792 | 0.1535 | 0.6372 | 0.2385 |
| **Instances** | | | | | | | | | | |
| 100 | 0.8148 | 0.1465 | 0.7912 | 0.1668 | 0.7836 | 0.1724 | 0.4333 | 0.1261 | 0.8182 | 0.1449 |
| 200 | 0.7701 | 0.1991 | 0.7475 | 0.2083 | 0.7418 | 0.2141 | 0.4408 | 0.1256 | 0.7788 | 0.1895 |
| 500 | 0.6221 | 0.2167 | 0.5843 | 0.2016 | 0.5764 | 0.2037 | 0.3864 | 0.1269 | 0.6390 | 0.2005 |
| 1000 | 0.5658 | 0.2672 | 0.5355 | 0.2536 | 0.5275 | 0.2568 | 0.3728 | 0.1617 | 0.5874 | 0.2461 |
| 2000 | 0.4495 | 0.2576 | 0.4252 | 0.2284 | 0.4193 | 0.2293 | 0.3143 | 0.1509 | 0.4779 | 0.2376 |
| **Attributes** | | | | | | | | | | |
| 30 | 0.6124 | 0.2653 | 0.6181 | 0.2371 | 0.6142 | 0.2382 | 0.4450 | 0.1462 | 0.6362 | 0.2383 |
| 50 | 0.6123 | 0.2651 | 0.6020 | 0.2457 | 0.5998 | 0.2477 | 0.4097 | 0.1485 | 0.6328 | 0.2432 |
| 100 | 0.6123 | 0.2659 | 0.5863 | 0.2540 | 0.5789 | 0.2559 | 0.3912 | 0.1312 | 0.6308 | 0.2455 |
| 150 | 0.6117 | 0.2653 | 0.5761 | 0.2585 | 0.5700 | 0.2594 | 0.3680 | 0.1507 | 0.6284 | 0.2479 |
| 200 | 0.6119 | 0.2642 | 0.5740 | 0.2601 | 0.5641 | 0.2636 | 0.3480 | 0.1526 | 0.6281 | 0.2497 |
| 250 | 0.6111 | 0.2657 | 0.5709 | 0.2613 | 0.5607 | 0.2631 | 0.3598 | 0.1414 | 0.6281 | 0.2494 |
| 1000 | 0.6120 | 0.2649 | 0.5581 | 0.2647 | 0.5485 | 0.2701 | 0.3365 | 0.1451 | 0.6255 | 0.2529 |
| **Classes** | | | | | | | | | | |
| 30 | 0.3133 | 0.2113 | 0.3177 | 0.1814 | 0.3088 | 0.1775 | 0.2360 | 0.0955 | 0.3563 | 0.1897 |
| 50 | 0.4645 | 0.2315 | 0.4464 | 0.2148 | 0.4366 | 0.2156 | 0.3004 | 0.1118 | 0.4922 | 0.2127 |
| 100 | 0.6668 | 0.2279 | 0.6343 | 0.2397 | 0.6270 | 0.2431 | 0.3910 | 0.1345 | 0.6800 | 0.2171 |
| 150 | 0.6912 | 0.1656 | 0.6467 | 0.1803 | 0.6407 | 0.1842 | 0.4183 | 0.0981 | 0.7003 | 0.1585 |
| 200 | 0.7621 | 0.1636 | 0.7203 | 0.1881 | 0.7146 | 0.1922 | 0.4550 | 0.1161 | 0.7690 | 0.1586 |
| 250 | 0.7369 | 0.1009 | 0.6791 | 0.1156 | 0.6725 | 0.1187 | 0.4457 | 0.1049 | 0.7431 | 0.0980 |
| 1000 | 0.9437 | 0.0560 | 0.9227 | 0.0782 | 0.9222 | 0.0787 | 0.5825 | 0.1413 | 0.9449 | 0.0548 |

The DataBank (https://databank.worldbank.org/, accessed June, 2023) also reports a multitude of yearly values of indicators related to various socio-economic statuses of countries around the world. Among them, we find Health, Nutrition, and Population statistics (https://databank.worldbank.org/source/health-nutrition-and-population-statistics, accessed June 2023). There are 467 indicators for 266 countries, for health, nutrition, and population. To illustrate the behaviour of MNC on real data, we have used the health, nutrition, and population indicators from 2021 to group countries. When retrieving the data from the database, we found that more values were available for this year than for the latest, 2022.

**Table 4  Values of the NMI and number of clusters obtained by each method for the real-world benchmarks.**

| Data | MN-C | /count | Kmeans | /count | GM | /count | AP | /count | Birch | /count |
|------|------|--------|--------|--------|------|--------|------|--------|-------|--------|
| R1 | 0.238 | 2 | 0.096 | 2 | 0.015 | 2 | 0.000 | 1 | 0.078 | 2 |
| R2 | 0.209 | 4 | 0.185 | 4 | 0.182 | 4 | 0.205 | 20 | 0.177 | 4 |
| R3 | 0.288 | 23 | 0.265 | 23 | 0.262 | 23 | 0.221 | 13 | 0.251 | 23 |
| R4 | 0.053 | 5 | 0.012 | 5 | 0.046 | 5 | 0.000 | 1 | 0.017 | 5 |
| R5 | 0.326 | 2 | 0.010 | 2 | 0.079 | 2 | 0.063 | 55 | 0.016 | 2 |
| R6 | 0.662 | 2 | 0.041 | 2 | 0.026 | 2 | 0.453 | 11 | 0.031 | 2 |
| R7 | 0.078 | 13 | 0.028 | 13 | 0.023 | 13 | 0.000 | 1 | 0.023 | 13 |
| R8 | 0.015 | 2 | 0.001 | 2 | 0.007 | 2 | 0.000 | 1 | 0.001 | 2 |
| R9 | 0.022 | 2 | 0.022 | 2 | 0.022 | 2 | 0.000 | 1 | 0.015 | 2 |
| R10 | 0.620 | 10 | 0.047 | 10 | 0.531 | 10 | 0.066 | 29 | 0.044 | 10 |
| R11 | 0.446 | 6 | 0.103 | 6 | 0.291 | 6 | 0.305 | 14 | 0.111 | 6 |
| R12 | 0.180 | 8 | 0.079 | 8 | 0.084 | 8 | 0.142 | 21 | 0.092 | 8 |
| R13 | 0.155 | 8 | 0.110 | 8 | 0.120 | 8 | 0.000 | 1 | 0.115 | 8 |
| R14 | 0.163 | 5 | 0.042 | 5 | 0.112 | 5 | 0.116 | 22 | 0.051 | 5 |
| R15 | 0.456 | 30 | 0.377 | 30 | 0.407 | 30 | 0.312 | 20 | 0.309 | 30 |
| R16 | 0.293 | 3 | 0.210 | 3 | 0.160 | 3 | 0.234 | 8 | 0.027 | 3 |

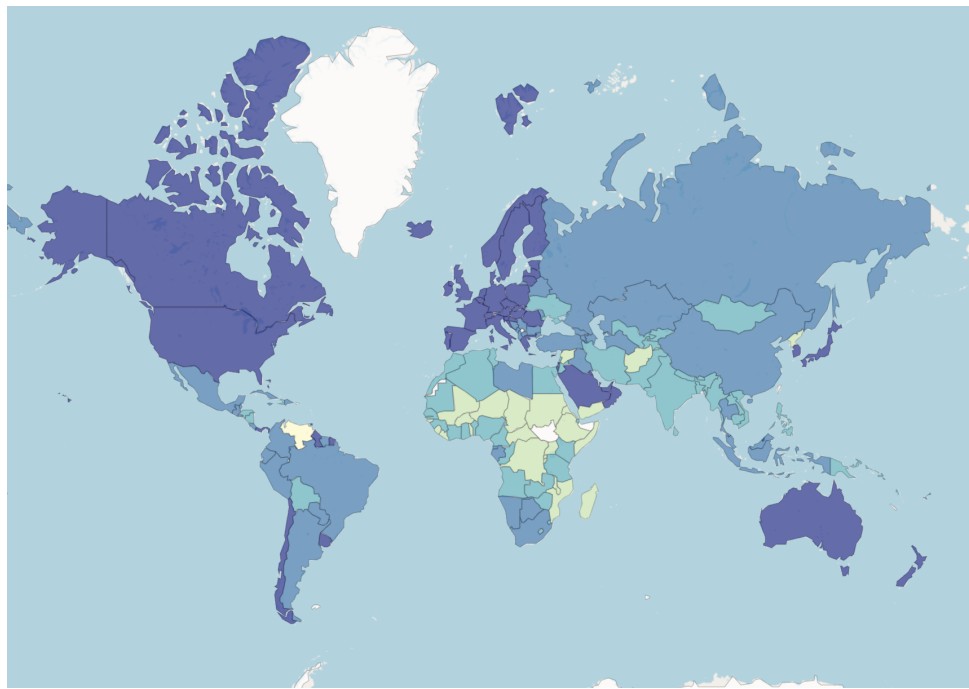

**Figure 5  World Bank data.** Countries are coloured based on their income group.

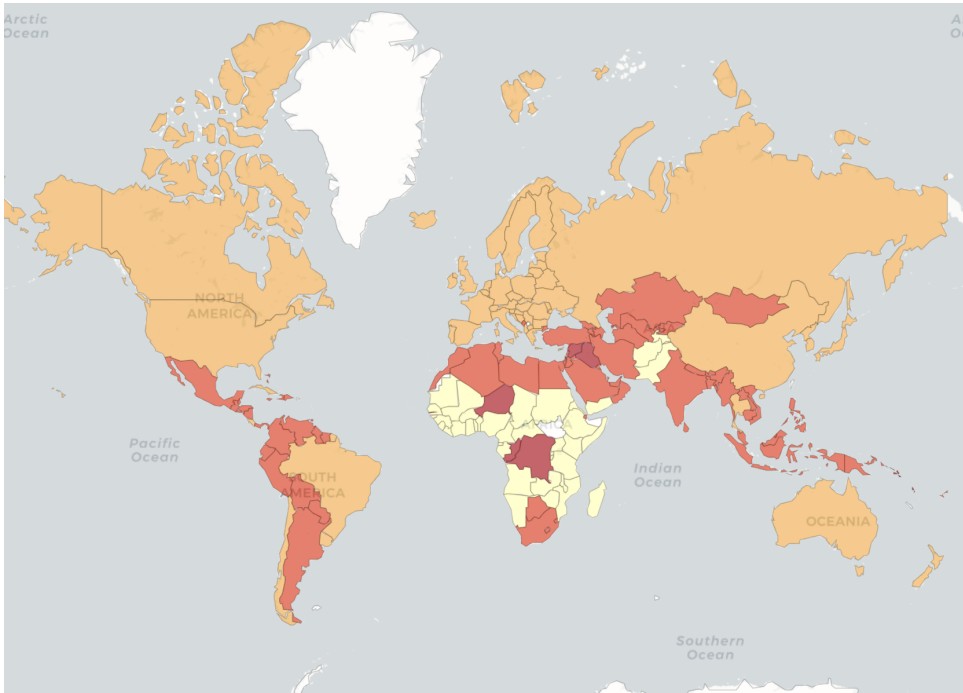

**Figure 6  World Bank data.** Countries are coloured based on MNC clustering results.

Because of the large number of missing values in the data, we discarded all indicators with more than one-half the number of countries. Similarly, we also discarded countries with more than 30 indicators with missing values. This resulted in a data set with 193 countries and 245 indicators. The rest of the missing values were replaced with the average value of that indicator for the corresponding region, which is part of the data set.

We ran all the algorithms on the resulting data set. MNC obtained an NMI of 0.32 for the four clusters. All the other methods obtained NMIs below 0.05. Figure 6 illustrates the four country groups identified by MNC. While these groups do not overlap with the income level classes, their distributions are not dissimilar. We find that most countries in the upper-middle and high income groups are in the same group in terms of the given health indicators. For example, while China and Russia are in a different income group than the United States of America and Canada (upper middle income *versus* high income), they are placed all together, also with high-income countries from Europe, by MNC based on the health indicators.

The clusters identified by MNC may also indicate that there is a common level for health, nutrition, and population status among countries with similar income as well as based on geographic regions. It also shows that lower-income countries have similar values for these indicators also.

# CONCLUSIONS

A simple network-based approach to the clustering problem has been presented. Data attributes are separated in intervals and placed in the nodes of a layer of a multipartite network. Thus, the network has a number of layers equal to the number of attributes. Each data instance adds a path to the network from the first to the last layer. MN-C identifies clusters by finding instances that are on the same or close paths in the network. Numerical experiments show that the approach is competitive against some standard state-of-the-art clustering techniques on a set of synthetic and real-world benchmarks. A real-world application that groups countries based on Health Nutrition and Population information available from the World Bank database is presented. Future research directions can explore different ways of constructing intervals for the network nodes and finding ways to identify or recommend a number of clusters.

## Funding

This work was supported by a grant of the Romanian Ministry of Education and Research, CNCS - UEFISCDI, project number PN-III-P4-ID-PCE-2020-2360, within PNCDI III. The funders had no role in study design, data collection and analysis, decision to publish, or preparation of the manuscript.

## Grant Disclosures

The following grant information was disclosed by the author:
The Romanian Ministry of Education and Research, CNCS - UEFISCDI, project number PN-III-P4-ID-PCE-2020-2360, within PNCDI III.

## Competing Interests

The authors declare there are no competing interests.

## Author Contributions

- Rodica-Ioana Lung conceived and designed the experiments, performed the experiments, analyzed the data, performed the computation work, prepared figures and/or tables, authored or reviewed drafts of the article, and approved the final draft.

## Data Availability

  The synthetically generated data, and the related code, are available in the Supplemental File.
  The real-world datasets are available in the Supplemental File and at the UCI Machine Learning Repository: https://archive.ics.uci.edu/datasets.

## Supplemental Information

Supplemental information for this article can be found online at http://dx.doi.org/10.7717/peerj-cs.1621#supplemental-information.

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
