# Peer review of "A new clustering method based on multipartite networks"

_PeerJ Computer Science, doi:10.7717/peerj-cs.1621_

## Round 0.1 · original submission · Major Revisions

The reviewers have substantial concerns about this manuscript. The authors should provide point-to-point responses to address all the concerns and provide a revised manuscript with the revised parts being marked in different color.

Reviewer 1 ·

Basic reporting

There are several areas that require improvement in the writing. For instance, the citation format on line 20 is inconsistent with the rest of the paper and should be revised for consistency. The content from line 31 to 33 is confusing and needs clarification. Additionally, the term "neuronal" in line 43 should be replaced with "neural" for accuracy. The phrasing of lines 56-58 can be improved for clarity and coherence. Furthermore, it is necessary to provide a clear definition of the MN-C methodology, including how the data is organized and the dimension of notation, spanning from line 78 to 87.

Regarding related work, it is recommended to incorporate and compare with more recent studies, such as the following:
Yin, Hao, et al. "Local higher-order graph clustering." Proceedings of the 23rd ACM SIGKDD International Conference on Knowledge Discovery and Data Mining. 2017.
Wang, Chun, et al. "Mgae: Marginalized graph autoencoder for graph clustering." Proceedings of the 2017 ACM on Conference on Information and Knowledge Management. 2017.
Tsitsulin, Anton, et al. "Graph clustering with graph neural networks." arXiv preprint arXiv:2006.16904 (2020).

Experimental design

Furthermore, Figure 2 lacks the definitions of diamond, square, and plus sign, which should be included for clarity and better understanding. Additionally, the results of the MN-C approach are missing from Table 1, and it is necessary to provide these results to complete the experimental findings and validate the proposed method. Besides, the author does not include enough details on how to repeat the experiment randomly which can be important since many clustering in the comparison are well known to be sensitive the random initial settings. Besides, the author should also provides the results of detailed comparison when the hyper-parameter such as cluster number varies.

Validity of the findings

While the question addressed in this research may not be entirely novel, the author presents a unique and innovative solution. In order to further strengthen the credibility and relevance of the proposed approach, it is highly recommended that the author seeks additional real-world applications as evidence for the necessity of this novel solution. Demonstrating the applicability of the proposed method in practical scenarios would significantly contribute to the scholarly contribution of the study.

Cite this review as

Reviewer 2 ·

Basic reporting

Clustering methods play a crucial role in uncovering hidden connections within data and have numerous practical applications. Traditional methods provide reliable and interpretable results, while modern approaches incorporate advanced techniques. The MN-C algorithm presents a novel approach to clustering, showcasing its behavior through experimental evaluations on different datasets.

Experimental design

There are several points the author should improve:
1. In the introduction section, the author states that there are multiple criteria for classifying clustering methods, but recent trends have grouped them into two major categories: traditional and modern approaches. To make this classification more comprehensive and understandable, it would be beneficial for the author to provide references that support this criterion. Including references to established literature on clustering methods and their classification would strengthen the rationale behind the classification and aid readers in understanding the categorization.


2. The author should carefully review the manuscript for small errors and inconsistencies. For example, in line 20, there is an issue with the reference format. In line 47, the abbreviation should be enclosed in parentheses. A corrected format would be: "Clustering Large Applications (CLARA)"

Validity of the findings

There are several points the author should improve:
1. It is advisable for the author to include more discussion sections throughout the paper. While the results and discussion section describes the findings, it lacks in-depth analysis and explanation of why the MN-C algorithm outperforms other methods. Expanding the discussion would enhance the clarity and completeness of the paper.
2.The author compares their proposed method with GM, Kmeans, AP, and Brich. However, it would be valuable to also compare the MN-C algorithm with other clustering methods based on neural networks, such as auto-encoders or other relevant techniques. Including a broader range of comparison methods would provide a more comprehensive evaluation of the MN-C algorithm's performance and highlight its advantages compared to various existing approaches.

Additional comments

The abstract should be rewritten to ensure it encompasses the necessary components. A well-written abstract typically includes the background, method, results, and discussion. Currently, the abstract focuses primarily on the background, with limited information provided on the other three aspects. Expanding the abstract to adequately summarize the method, key results, and important discussion points would provide readers with a concise overview of the paper's contributions and findings.

Cite this review as

·

Basic reporting

In this paper, the author proposes a new graph-based clustering method by constructing multipartite networks and then grouping paths connecting all layers. Using simulation and real-world datasets, the author benchmarked this method with other four existing methods to demonstrate its utility. Generally, I found this approach reasonable and interesting. Below are some specific comments and suggestions to strengthen this work.

1) This method is overall heuristic without theoretical proof, which is acceptable. However, its unique advantage has not been clearly discussed in this paper. It would be helpful if the author could provide a discussion on the scenarios in which people would expect this method performs better than other clustering methods.

Experimental design

2) Given this approach belongs to graph-based (or network-based) clustering methods in the general sense, I would also expect to see the comparison with some widely-applied graph methods, e.g., spectral clustering and Louvain method.

Validity of the findings

3) One re-occurring problem in clustering methodologies is that there is usually a lack of exposition on how the clustering resolution should be determined, given the users may not have the prior knowledge of what k would indicate the optimal number of clusters. Hence, it would be helpful if the author provides a robustness evaluation on how different selection of k affects the clustering performance. Or alternatively, is there a way to automatically find k based on an optimization scheme (e.g., the elbow-method used in K-means clustering)?

Additional comments

4) For each layer corresponding to each attribute in the data, the author assigns the nodes into equal-size intervals. There is a caveat that once the data (in terms of a single attribute) has non-uniform density in its distribution, this may lead to fragmentation of a dense cluster. Can any density-based one-dimensional clustering algorithm contribute to a better performance than uniformly cutting the layer into equal-sized bins?

5) I thank the author for providing the scripts used for the analyses in this paper.

Cite this review as

---

## Round 0.2 · accepted · Accept

All reviewers are satisfied with the revisions and I concur to recommend accepting this manuscript.

Reviewer 1 ·

Basic reporting

In reviewing this article, I observed a robust grounding in relevant literature and a thorough provision of field context. The manuscript showcases clarity in its professional structure, supported by figures, tables, and accessible raw data. Moreover, it commendably offers detailed explanations, clear term definitions, and comprehensive proofs corresponding to the presented theorems.

Experimental design

The authors present original research fitting the journal's Aims and Scope, addressing a distinct knowledge gap with a well-defined and meaningful question. The study stands out for its rigorous approach and provides a detailed methodology ensuring replicability and upholding ethical standards.

Validity of the findings

While I did not assess impact and novelty, I commend the manuscript for its clear rationale and encouragement of meaningful replication that benefits the literature. The provided data is robust and statistically sound, with conclusions aptly linked to the original research question and strictly based on the supporting results.

Cite this review as

Reviewer 2 ·

Basic reporting

All satisfied.

Experimental design

All satisfied.

Validity of the findings

All satisfied.

Additional comments

All satisfied.

Cite this review as

·

Basic reporting

no comment

Experimental design

no comment

Validity of the findings

no comment

Additional comments

Overall, I’m pleased to see that the authors have thoroughly revised the manuscript to reflect these points and addressed these questions nicely. I don’t have additional specific questions for the authors.

Cite this review as